# The Endothelial Transcription Factor ERG Mediates a Differential Role in the Aneurysmatic Ascending Aorta with Bicuspid or Tricuspid Aorta Valve: A Preliminary Study

**DOI:** 10.3390/ijms231810848

**Published:** 2022-09-16

**Authors:** Calogera Pisano, Sonia Terriaca, Maria Giovanna Scioli, Paolo Nardi, Claudia Altieri, Augusto Orlandi, Giovanni Ruvolo, Carmela Rita Balistreri

**Affiliations:** 1Department of Cardiac Surgery, Tor Vergata University Polyclinic, 00133 Rome, Italy; 2Pathological Anatomy, Department of Biomedicine and Prevention, Tor Vergata University, 00133 Rome, Italy; 3Department of Biomedical Sciences, Catholic University of Our Lady of Good Counsel, 1001 Tirana, Albania; 4Cellular and Molecular Laboratory, Department of Biomedicine, Neuroscience and Advanced Diagnostics (Bi.N.D.), University of Palermo, 90134 Palermo, Italy

**Keywords:** ascending aorta aneurysm, bicuspid aorta valve, tricuspid aorta valve, ERG transcriptional factor pathway, TGF-β-SMAD, Notch, NO pathways modulation

## Abstract

The pathobiology of ascending aorta aneurysms (AAA) onset and progression is not well understood and only partially characterized. AAA are also complicated in case of bicuspid aorta valve (BAV) anatomy. There is emerging evidence about the crucial role of endothelium-related pathways, which show in AAA an altered expression and function. Here, we examined the involvement of ERG-related pathways in the differential progression of disease in aortic tissues from patients having a BAV or tricuspid aorta valve (TAV) with or without AAA. Our findings identified ERG as a novel endothelial-specific regulator of TGF-β-SMAD, Notch, and NO pathways, by modulating a differential fibrotic or calcified AAA progression in BAV and TAV aortas. We provided evidence that calcification is correlated to different ERG expression (as gene and protein), which appears to be under control of Notch signaling. The latter, when increased, associated with an early calcification in aortas with BAV valve and aneurysmatic, was demonstrated to favor the progression versus severe complications, i.e., dissection or rupture. In TAV aneurysmatic aortas, ERG appeared to modulate fibrosis. Therefore, we proposed that ERG may represent a sensitive tissue biomarker to monitor AAA progression and a target to develop therapeutic strategies and influence surgical procedures.

## 1. Introduction

Current evidence underlines that ascending aorta aneurysm (AAA) is not the result of a unique risk factor, but rather appears as a multifactorial disease, having a heterogenous and complex cluster of cardiovascular abnormalities at diverse levels, from the genetic and epigenetic to molecular and cellular levels [1,2,3]. However, AAA’s biology of onset and progression is yet not well understood, and remains partially characterized, and particularly for the sporadic forms [2,3]. Such disease is also complicated in case of bicuspid aorta valve (BAV) anatomy [4,5,6]. AAA in BAV cases appears to be characterized by a typical pathogenesis and an early onset’s age, as well as by an increased risk of evolving in severe aortic complications (i.e., dissection or rupture), when certain diameters, as well as a significant rate of calcification, are reached [4,5,6]. Currently, it is emerging the crucial role of endothelial dysfunction in AAA development and dissection/rupture associated with BAV, closely related to deregulation of regulatory pathways principally in endothelial cells (EC), and consequently in vascular smooth muscle cells (VSMC) [4,5,6]. Consistent with this, we have demonstrated in BAV cases, when compared with subjects with tricuspid aortic valve (TAV), a deregulation of crucial EC pathways (i.e., Notch, TGF-β, TLR-4 and others), a reduced capacity of vascular repair significantly related to a decreased blood number of endothelial progenitor cells (EPC), and an altered T and B immune response to tissue damage [7,8,9,10,11]. In addition, we have also demonstrated that such endothelial alterations in BAV are significantly associated with an increased VSMC apoptosis, as well as a wall-remodeling characterized by calcification. We have also reported that such mechanisms positively correlate to an earlier progression of BAV versus moderate (i.e., aorta stenosis and calcification) or severe AAA complications, including dissection and rupture [7,8,9,10,11]. However, it is not clear which of the abovementioned mechanisms, or ones not known, are the main contributors, of aorta BAV complications, i.e., AAA. In addition, it is not entirely recognized whether hemodynamic and environmental factors also play a fundamental role, as well as epigenetic or transcriptional factors can act as hub or be result of a complex interplay of pathway networks [10,12]. Probably, a synergistic action of all these elements reflects the different AAA phenotypes associated with BAV, as stressed in our works, than those detected in TAV individuals [4,10,12]. Differently, TAV subjects show a more advanced age of AAA onset, about versus the 70–75 years, and a pathogenesis more related to vascular aging and the resultant remodeling and degeneration process associated with a preeminent fibrosis, that significantly reduces the probability of AAA progression in dissection and rupture [2,13,14,15,16,17]. Precisely, a typical vascular remodeling and degeneration, accompanied by wound healing associated with a significant increase of circulating EPC levels [9,11], tissue expression of TGF-β and Smad-3 [18,19,20,21] consequent inflammation, endothelial-to-mesenchymal transition (EndMT) [22,23,24] and fibrosis [25], embody aorta dilation, or better, AAA disease, in TAV patients.

These observations lead to increasingly strengthen the hypothesis that AAA in aortas with TAV or BAV valves uses different trajectories both in development and progression, where the differential mechanisms involved requires to be fully elucidated, by detecting all the molecules involved, and particularly the hub pathway, having the role of principal driver and regulator of other crucial pathways. Consequently, diverse aspects remain unresolved and numerous questions surface, even if the interpretation of current evidence suggests the crucial role of endothelium and its pathways, which show an expression and functions dependent on the maintained EC homeostasis or its alterations [14,26,27,28,29]. Such leads us to investigate on the transcriptional modulation of aorta endothelium in conditions of BAV or TAV, with or without AAA. Accordingly, a genome-wide association study [30] and the works of Dr Randi [31,32,33,34,35] have evidenced the fundamental role in EC of a transcription factor, ERG, encoded by *ETS-related* gene, able to regulate endothelial homeostasis and the consequent vascular stability, as well as its alterations, and the evocation of pathological conditions, by controlling a wide range of targets and pathways, including Notch, TLR-4 and TGF-β-SMAD1/3 [31,32,33,34,35,36]. Precisely, here, we assessed the eventual involvement of ERG factor in aortas from BAV and TAV cases with or without AAA, and in the mechanisms related to is differential progression. Such investigation was conducted a broad basis to first get a complete identification on the potential differences in expression and roles among the diverse groups (four: with or without AAA, respectively with aorta diameter ≥45 mm or ≤45 mm, see guidelines indicated in reference [37]) of BAV and TAV cases included in the study. To achieve such main aim, our study was complemented by histological and immunohistochemical investigations on tissue aorta samples from four groups (BAV, BAV with AAA, TAV and TAV with AAA), as well as by an extensive real time PCR-based gene expression analyses. The goal of this study was to reveal new, yet unidentified, alterations in EC from aneurysmatic or not aortas, by comparing BAV versus TAV EC, as well as to diversify the roles of specific detected molecules in AAA formation and progression. Based on the results obtained, we also evidenced a potential model of progression of AAA.

## 2. Results

### 2.1. BAV and TAV Patient Characteristics

Demographic and clinical characteristics of the study population are synthesized in Table 1. BAV and TAV patient’s features are summarized in Table 2. No significant differences were observed regarding the size of aorta dilatation between BAV and TAV cases affected by AAA. However, BAV cases are younger than TAV cases. Hypertension was a common risk factor to both BAV and TAV patients, although in BAV population the percentage of hypertensive was higher. Renal failure and diabetes were more frequent in TAV patients than in BAV. The mean aortic root diameter in BAV patients was 42.6 ± 6.1 mm, instead in TAV patients was 41.9 ± 5.6 mm. The mean ascending aorta diameter in BAV patients was 50.2 ± 6.8 mm, instead in TAV patients was 52.7 ± 8.9 mm. As regards the morphological aspects detectable during the cardiac operation, in 100% of BAV population we detected a coronary ostia dislocation and an origin of the epi-aortic vessels from the convexity of the ascending aorta, in 10% of cases a Valsalva sinus prolapse, in 30% an asymmetric dilatation of the ascending aorta and in 80% an aortic wall thickness. In TAV patients, we revealed a coronary ostia dislocation in 90% of patients, an asymmetric dilatation of the ascending aorta in 30% of patients, a left ventricle/aortic valve disjunction in 20% of cases, an aortic wall thickness in 60% of cases. Comparing TAV and BAV patients with an ascending aorta diameter ≥45 mm and >45 mm (Table 3), we noticed that the Valsalva sinus prolapse was significantly associated to TAV patients with an ascending aorta diameter <45 (*Group 4*), instead the asymmetric dilatation of the ascending aorta was significantly associated to BAV and TAV patients with an ascending aorta <45 mm (*Group 2* and *Group 4*).

### 2.2. Differential Expression of Endothelial ERG Transcription Factor in BAV vs. TAV Aortic Intima

In our study, immunohistochemical and gene expression analyses evidenced that aortic tissues of BAV cases with AAA had a significantly higher percentage of ERG^+^ EC cells in their aortic tissue samples, accompanied by a significant difference in gene *ERG* transcription than the other groups (see Figure 1). By contrast, a significant reduced number of aortic intimal ERG^+^ EC cells, as well as a significant decreased level of ERG gene characterized TAV cases with or without AAA (see Figure 1).

### 2.3. Upregulation of Tissue ERG Gene Expression in BAV Cases with AAA Correlates with miR126 Levels

Since recent literature reports evidence that miR-126-5P targets mesenchymal genes (i.e., *SMAD3/2* genes) [36] in EC by inhibiting EndMT transition, we evaluated its expression in aorta intimal tissue extracts. Interestingly, we observed that miR-126-5P was significantly up-expressed only in tissue aneurysmatic samples from BAV cases (see Figure 1), and it correlates positively with *ERG* gene levels (r = +0.29, *p* = 0.02, by linear logistic regression, Pearson’s test).

### 2.4. Downregulation of ERG Gene and miR126 Reciprocally Promotes Higher Expression of SMAD3 in TAV Aortic Tissues with AAA and Higher Levels of αSMA^+^/S100A4^+^ EC and EndMTs

To shed light on the mechanisms promoted by *ERG* gene and *miR-126-5P* downregulation in intimal aortic tissue of TAV, we performed immunohistochemistry. We detected higher levels of SMAD3 and αSMA^+^/S100A4^+^ molecules, in the aortic EC from the intimal samples of the TAV group with AAA than in other aorta tissue samples of other groups. Such results suggested an increased aorta fibrosis in aorta aneurismatic TAV tissues, being the molecules identified the typical mesenchymal biomarkers of EndMT transition (see Figure 2 and Figure 3 and related Table 3).

### 2.5. A Higher Rate of Fibrosis Characterizes TAV Aortic Tissues with AAA, as Well as Increased Calcification in BAV Tissues with AAA

To validate the results obtained, we evaluated the rate of fibrosis in the aorta tissues by using Masson’s trichrome staining (Table 3 and Figure 4). Remarkably, we observed the highest grade of fibrosis in aortic medial tissue samples from TAV cases with AAA than in other groups (see Figure 4). By using Alizarin red staining, we also detected a higher degree of calcification in BAV than TAV aortic tissues with AAA (Figure 4).

### 2.6. Decreased Levels of Notch Intracellular Domain (NICD) in EC and VSMCs from BAV vs. TAV Aorta Tissues

We also assessed by immunohistochemistry the percentage of Notch intracellular domain (NICD)^+^ EC and VSMCs in BAV vs. TAV aortic tissues. Interestingly, we observed significantly reduced percentages of both NICD^+^ EC and VSMCs in BAV vs. TAV aorta tissues, with more reduced values in BAV cases with non-aneurysmatic aortic tissues (see Figure 3 and Figure 4), by confirming the deregulated expression of Notch pathway in BAV cases. However, the comparison of obtained data led us to evidence an interesting aspect, that is a positive correlation between NICD^+^ and ERG^+^ EC in BAV aortic tissues from AAA or non-aneurysmatic groups (r = +0.17 and r = +0.29, *p* = 0.02 and *p* = 0.003, respectively, by linear logistic regression, Pearson test). Precisely, we observed comparable reduced percentages of NICD^+^ and ERG^+^ EC in aorta BAV tissues from non-anurysmatic group, and a similar increase of NICD^+^ and ERG^+^ EC in BAV aorta with AAA, by likely suggesting a close relationship between the two molecules. Differently, we detected the highest levels of NICD^+^ EC and VSMCs in aorta tissues from non and aneurysmatic TAV groups. Furthermore, such higher levels correlate with the significantly reduced calcification observed in aorta TAV than BAV tissues (r = −0.25, *p* = 0.04, by linear logistic regression, Pearson’s test, Figure 4).

### 2.7. Higher eNOS Levels in BAV vs. TAV Aortic Tissues

To provide further data for elucidating the different rate of calcification observed in BAV aorta tissues, we also completed our appraisals by detecting the percentages of eNOS^+^ EC in BAV vs. TAV aorta tissues by using immunohistochemical analysis. Interestingly, we observed a higher number of eNOS^+^ EC in all the BAV aorta tissues, with a trend in amplitude in aneurysmatic aorta tissues (see Figure 2).

### 2.8. Upregulated Expression of ERG Endothelial Transcription Factor and miR-126-5P in Aortic Medial Tissues

Given the interesting data obtained in examining aorta intima tissues, we also detected the levels of gene expression of *ERG* in aorta media tissues, by evidencing significative values in BAV with AAA tissue samples than the other aorta media tissue examined (Figure 4).

## 3. Discussion

Our histological and immunohistochemical investigations conducted in tissue aorta samples from four groups of individuals enrolled in study (BAV, BAV with AAA, TAV and TAV with AAA), and cotemporally complemented by extensive real time PCR-based gene expression analyses, permitted to obtain promising findings, which agree our suggestion about AAA in aortas with a BAV or TAV valve of following different trajectories both in development and progression. Precisely, they suggest that ERG transcriptional factor drives a differential transcription and expression of molecular pathways correlated to a diverse aorta wall remodeling and degeneration in case of BAV or TAV condition. Such, indeed, results in the evocation of diverse pathological conditions, ranging from EndMT transition to higher fibrosis, or from high levels of miR-126-5P to higher calcification, by controlling a wide range of targets and pathways, including Notch, SMAD3 and αSMA^+^/S100A4^+^ molecules. Accordingly, we observed in aortic tissues of BAV cases with AAA a significantly higher percentage of ERG^+^ EC cells, accompanied by a significant difference transcriptional levels of *ERG* gene than the other groups. Differently, TAV cases with or without AAA showed a significantly reduced number of aortic intimal ERG^+^ EC cells, as well as a significantly decreased level of *ERG* gene transcription. In addition, we also detected in tissue aneurysmatic samples from BAV cases an up-expression of miR-126-5P, known targeted mesenchymal genes (i.e., *SMAD3/2* genes) [36] in EC by inhibiting EndMT transition, that positively correlated with higher *ERG* gene transcription levels. Consistent with such data, we assessed by immunohistochemistry analysis and via Masson’s trichrome staining, significantly decreased levels of SMAD3 and αSMA^+^/S100A4^+^ molecules and a significantly reduced rate of fibrosis in tissue aorta samples from BAV cases with AAA when compared with TAV cases. Furthermore, an increased degree of calcification in BAV with AAA than TAV aortic tissues with AAA was observed by using Alizarin red staining. In contrast, we detected, in aortic medial tissue samples from TAV cases with AAA than the other groups, the highest grade of fibrosis associated with the highest levels of SMAD3 and αSMA^+^/S100A4^+^ molecules, the typical mesenchymal biomarkers of EndMT transition.

The interpretation of such findings obtained led us to suggest that ERG likely regulates the gene expression of miR-126-5P, which in turn modulates the gene expression of SMAD molecules, by especially inducing their inhibition. This would explain the reduced collagen accumulation and fibrosis in all the areas of aneurysmatic BAV aortic tissues, where an increased Alizarin red specific staining was while evidenced by indicating an increased calcification process.

In the literature, our data are in accordance with the results obtained by the group of Randi, that have recently demonstrated the ERG effect on canonical TGF-β-SMAD signaling [31,32,33,34,35]. They have shown that ablation of ERG expression prevents vascular fibrosis, due to the inhibition of SMAD3 activity and activation of SMAD1 pathway. The inhibition of ERG expression was evaluated by Randi’s group in liver chronic diseases and correlated with EndMT and the rate of SMAD 3-mediated fibrogenesis. Likewise, Zhang and coworkers [38] have reported that ERG reduces cardiac fibrosis via the inhibition of endthelin-1, and Nagai group [36] has evidenced that the downregulation of ERG in EC triggers EndMT.

Furthermore, we also showed that BAV tissues had significantly reduced percentages of both NICD^+^ EC and VSMCs than TAV aorta tissues, with more reduced values observed in BAV cases with non-aneurysmatic aortic tissues. Such datum has first confirmed the deregulated expression of Notch pathway that characterizes the BAV cases. Besides, the comparison of obtained data led us to evidence a positive correlation between NICD^+^ and ERG^+^ EC in BAV aortic tissues from AAA or non-aneurysmatic groups. Precisely, we detected comparable reduced percentages of NICD^+^ and ERG^+^ EC in aorta BAV tissues from non-anurysmatic group, and a similar increase of NICD^+^ and ERG^+^ EC in BAV aorta with AAA, by likely suggesting a close relationship between the two molecules. Differently, we detected the highest levels of NICD^+^ EC and VSMCs in aorta tissues from non and aneurysmatic TAV groups. Current evidence likely led us to suppose that Notch signaling regulates ERG expression by promoting a positive ERG-Notch loop, which in turn controls the expression of other key pathways crucial in AAA progression [39,40,41]. We hypothesized that such interplay can likely determine a gene profile responsible of calcification, as well as of BAV severe complications observed in aneurysmatic aortic tissues. In contrast, we showed the highest levels of NICD^+^ EC and VSMCs in aortic tissues from aneurysmatic TAV group. The latter might confirm the role of Notch signaling in triggering EndMT fibrosis, by activating BMP and TGF-β signaling, which in turn synergizes with Notch pathway to accelerate EndMT. Consistently, Notch signaling has been demonstrated both to inhibit runt-related transcription factor 2, a regulator of osteoblast cell fate, and to increase the expression of SRY-Box transcription factor 9, which is able to prevent calcification, as well as to evocate regulation of bone morphogenetic protein 2, also partially implicated in the calcification prevention [42,43,44,45,46,47,48,49].

Finally, we concluded our investigations by detecting the percentages of eNOS^+^ EC in BAV vs. TAV aorta tissues by using immunohistochemical analysis in order to further elucidate the different rate of calcification observed in BAV aorta tissues. Interestingly, we observed a higher number of eNOS^+^ EC in all the BAV aorta tissues, with a trend in amplitude in aneurysmatic aorta tissues. Such datum led us to hypothesize that this typical augment might be associated with an increased accumulation of NO uncoupled products contributing to the oxidative stress, and favoring early AAA progression toward adverse clinical events such as dissection or rupture [50]. Superoxide radicals can, indeed, react with NO, forming peroxynitrite, a potent oxidant, which can contribute to endothelial dysfunction, as well as to reduce NO bioavailability and. in turn, facilitate calcium deposition [50].

## 4. Materials and Methods 

### 4.1. Population Enrolled 

For this study, aortic samples were collected from the aortas of TAV (n = 5) and BAV (n = 5) cases with valve alterations and with ascending aorta diameter ≤45 mm and from TAV (n = 5) and BAV (n = 5) cases with ascending aorta diameter ≥45 mm undergone to elective surgical procedures between 2018 and 2022 in the Department of Cardio-surgery of Tor Vergata University of Rome. Appropriate exclusion criteria were also used during the BAV/TAV enrollment, for the following diseases: (a) cardiovascular diseases were excluded according to history and by detecting apposite laboratory and imaging biomarkers as indicated by more recent ESC or ASC guidelines; (b) connective tissue disorders were excluded by assessing markers of inflammation immunological (i.e., autoantibodies) and imaging biomarkers; (c) inflammatory diseases (from infections to hematological, gastrointestinal, urogenital, pulmonary, neurological, endocrinal inflammatory disorders, and tumors included) by detecting apposite laboratory (including complete blood cell count, erythrocyte sedimentation rate, glucose, urea nitrogen, creatinine, electrolytes, C reactive protein, liver function tests, iron, and proteins) and imaging biomarkers. In order to exclude ascending aorta aneurysms related to connective tissue syndromes (Marfan Syndrome, Ehlers Danlos, Loeys Diets) all patients undergo clinical (Ghent Criteria) and genetic screening in our “Reference Center of Rare Diseases and Marfan Syndrome” in Tor Vergata University. In addition, all the cases enrolled belonged to the same ethnic group. Thus, a very homogenous population was studied.

Surgical indication were: (1) aortic diameter ≥ 45 mm in presence of a severe TAV or BAV aortic valve dysfunction; (2) aortic diameter ≥ 50 mm in BAV patients without aortic valve dysfunction; (3) aortic diameter ≥ 55 in TAV patients without aortic valve dysfunction; (4) intraoperative findings unless the diameter: significant coronary ostia dislocation, aortic wall thickness, left ventricle/aortic valve disjunction with evidence of cardiac muscle in transparency at the level of the right/non coronary sinus, asymmetric dilatation of Valsalva sinus/sinuses [51]. In the grouping of patients, we have chosen a cut-off diameter of 45 mm because it is the diameter advocated in the “2021 ESC/EACTS Guidelines for the Management of Valvular Heart Disease” for ascending aorta replacement in patients with aortic valve diseases (severe stenosis or regurgitation). Patients with an ascending aorta < 45 mm underwent ascending aorta replacement only in presence of specific intraoperative findings indicated above. Furthermore, elective or acute surgical treatment (using wheat operation, Bentall-De Bono and Tirone David surgical techniques, whenever possible) and complementary tubular-ascending aorta resection were performed in the BAV and TAV patients with AAA after evaluation of aortic transverse diameter sizes by Computed Tomography scanning according to recent guidelines. Accordingly, an experienced physician evaluated aortic transverse diameter sizes by echocardiography (Philips iE. 33) before either elective or urgent surgery. The dimension of the aortic annulus, sinuses of Valsalva, proximal ascending aorta (above 2.5 cm of the sino-tubular junction) and aortic arch were assessed pre-operatively by trans-thoracic echocardiography as well as in the operating theatre by trans-oesophageal-echocardiography before the institution of the cardiopulmonary by-pass. These measures, together with demographic and clinical data (including comorbidities) were obtained from patients’ medical records and are presented in Table 1.

The Local Ethics Committee approved the study (protocol n.179/18-01-Aorta-2018) and written informed consent was obtained from each patient prior to the study. This study conformed to the principles outlined in the Declaration of Helsinki.

### 4.2. Histochemical and Immunohistochemical Analysis

Serial 4-µm thick paraffin sections from 10% neutral-buffered formalin-fixed aortic tissue samples were stained with Alizarin red and Masson’s trichrome staining. Calcium deposits and collagen fiber accumulation were evaluated calculating the fibrotic or calcified area through Image J software (NIH, Bethesda, MD, USA). Images were previously captured using a digital camera (DXM1200F, Nikon, Tokyo, Japan) connected to the ACT-1 software (NIH, Bethesda, MD, USA) [52] For immunohistochemistry, sections were reacted with rabbit monoclonal anti-phosphoSMAD3 (1:200; Abcam, Cambridge, UK), anti-ERG (1:150; Merck KGaA, Darmstadt, Germany) and anti-S100A4 (1:250; Dako, Santa Clara, CA, USA), rabbit polyclonal anti-Notch Intracellular Domain (NICD; 1:200; Bioss Antibodies Inc., Woburn, MA, USA), mouse monoclonal anti-αSMA (1:150; Dako) and anti-eNOS (1:100; BD Bioscences, Becton, Dickinson, UK). Positive and negative controls were used [52]. For the immunohistochemical evaluation, the percentage of positive endothelial cells/field (40× magnification) were counted.

### 4.3. Gene Expression Analysis

Total RNA and miRNA extraction from endothelium and tunica media derived from cryopreserved aorta samples was performed using the TRI Reagent^®^ (Sigma Aldrich, St. Louis, MO, USA) and the mirVana miRNA Isolation Kit (Thermo Fisher Scientific, Waltham, MA, USA) according to manufacturer’s protocols, respectively. 700 ng RNA were reverse transcribed using the MystiCq microRNA cDNA Synthesis Mix kit (Sigma Aldrich, St. Louis, MO, USA) or the SuperScript III (Invitrogen, Thermo Fisher Scientific, MA, USA). Real-time PCR was carried out by using SYBR Green (BioRad, Hercules, CA, USA). Precisely, to analyze the expression of miR-126-5P and ERG, we used MystiCq microRNA Primer HSA-miR-126-5P (Merck KGaA, Darmstadt, Germany) and MystiCq Universal PCR Primer (Merck KGaA, Darmstadt, Germany), ERG primer forward 5′-GGAGTGGGCGGTGAAAGA-3′ and ERG primer reverse 5′-AAGGATGTCGGCGTTGTAGC-3′ (Sigma Aldrich, St. Louis, MO, USA), respectively. Real-time PCR was performed on Applied Biosystem StepOnePLus (Thermofisher Scientific, Waltham, MA, USA). HSA-miR-126-5P was normalized to the expression of U6, while ERG was normalized to the expression of GAPDH. Changes in target gene expression levels were calculated using the comparative ΔΔCT method. Fold change was considered significant for values ≥2.0 and ≤0.5.

### 4.4. Statistical Analysis

Statistical analyses were performed using STATA software version 20. Significant differences among qualitative variables were calculated by using Fisher test. Continuous variables (including systemic blood molecule, protein, and gene expression levels) were expressed as mean ± SD (Standard deviation). Unpaired *t*-test (Welch corrected) was utilized to analyze the data between two groups, while one-way ANOVA or Kruskal-Wallis ‘s test followed by Bonferroni correction or Dunn test was applied to compare more than two groups. To identify possible correlations, a non-parametrical Spearman correlation test was also used, as well as a linear logistic regression using Pearson test. Differences were considered significant when a *p* value < 0.05 was obtained by comparison between the different groups.

## 5. Conclusions

In conclusion, our results strongly support ERG as a novel regulator of EC-specific targets and pathways, including canonical TGF-β-SMAD, Notch, and NO pathways, by modulating a differential progression (i.e., fibrotic or calcified) of AAA in BAV and TAV aortas. We provided evidence that levels of ERG expression (as gene and protein), particularly in aortic EC, appear to be in close relationship with Notch signaling and, when increased, can induce early calcification in aortas with BAV valve and aneurysmatic, favoring the progression towards dissection or rupture. In TAV aneurysmatic aortas, ERG appears to modulate fibrogenesis. This finding is in accordance to the specific function of ERG factor in regulating a complex transcriptional pathway involved in the development and function of endothelium, and consequently in the maintenance of vascular integrity of cardiovascular system, during both embryological and adult life, as well as in evocating disease. ERG knockout mice demonstrated to have an altered vascular development, incompatible with the life at E10.5–12.5 [31]. In addition, such defect is related to activation and stability of β-catenin/Wnt pathway [32].

Therefore, we propose that ERG may represent a sensitive tissue biomarker to monitor AAA progression and a target to develop therapeutic strategies and surgical procedures. We also suggest a model of context-specific combinatorial networks, that integrates this transcriptional factor with the pathways abovementioned in the onset and progression of aneurysmatic aorta with BAV or TAV, and studied in others recent studies conducted by our group [9,11,53] (Figure 5).

## 6. Limitations

Our study may be defined as a preliminary investigation, given the reduced number of patients enrolled. However, both AAA, even if in increase in old populations, and BAV syndrome are rare pathologies [4], and consequently the relative limited number of cases examined provides important evidence to develop on large numbers, and preferentially performing multicentered studies.

## Figures and Tables

**Figure 1 ijms-23-10848-f001:**
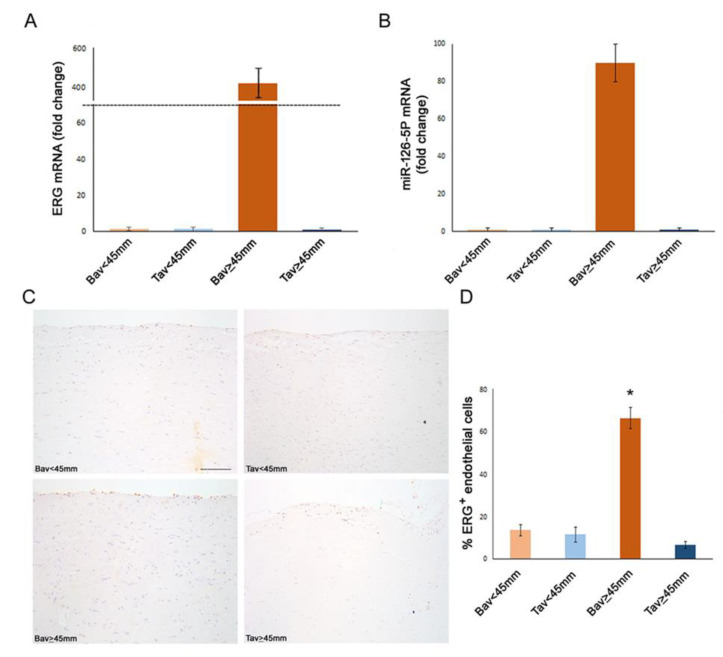
Increased ERG expression and miR-126-5P up-regulation in BAV tunica intima with aortic diameter ≥45 mm. Gene expression analysis on RNA extracted from tunica intima of aortic tissue show an increased level of ERG (**A**) and miR-126-5P (**B**) transcripts in BAV patients with aortic diameter ≥45 mm compared with the other groups. Changes in gene expression were calculated using the comparative ΔΔCT method. Fold change is considered significant for values >2.0 and <0.5. Representative images (**C**) and semiquantitative evaluation (**D**) of ERG immunostainings of BAV and TAV tunica intima that show an increased expression in BAV with aortic diameter ≥45 mm compared with the other groups. Averages are reported as percentage of ERG^+^ endothelial cells ± SEM. * *p* < 0.01. Scale bar = 10.

**Figure 2 ijms-23-10848-f002:**
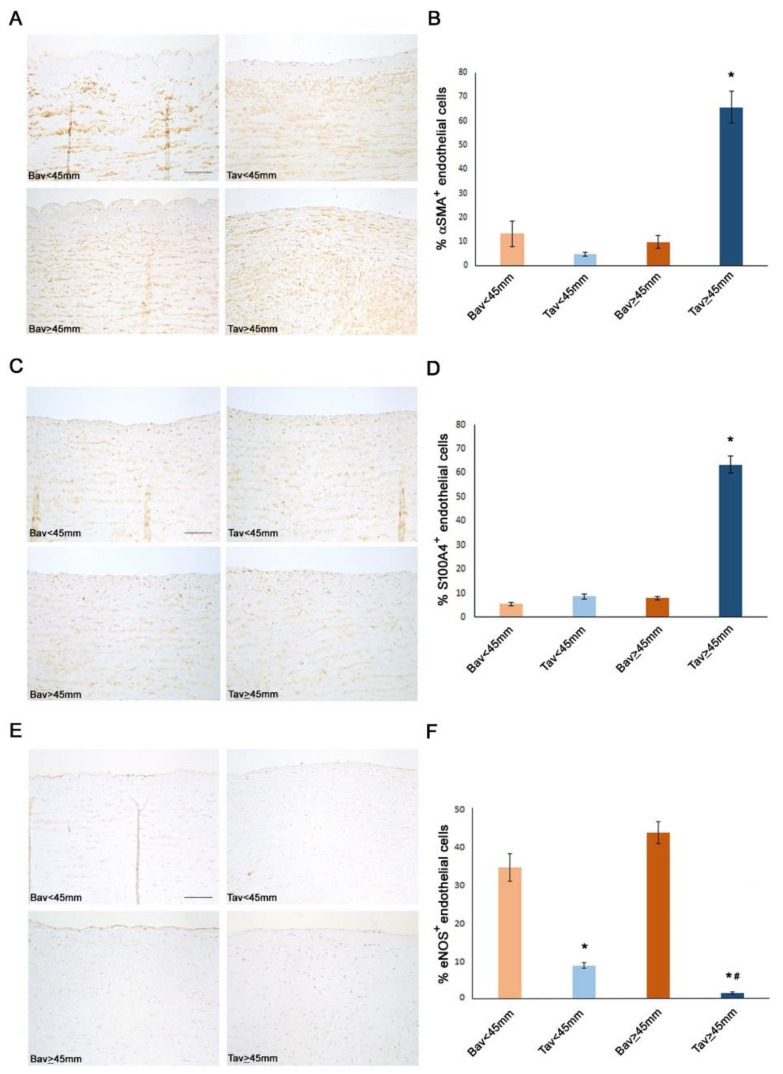
Higher levels of mesenchymal markers characterize TAV tunica intima with aortic diameter ≥ 45 mm. Representative images (**A**) and semiquantitative evaluation (**B**) of α-SMA immunostainings of BAV and TAV tunica intima show an increased expression in TAV with aortic diameter ≥45 mm compared with the other groups. Scale bar = 100 µm. Representative images (**C**) and semiquantitative evaluation (**D**) of S100A4 immunostainings of BAV and TAV tunica intima show an increased expression in TAV with aortic diameter ≥ 45 mm compared with the other groups. Scale bar = 100 µm. Representative images (**E**) and semiquantitative evaluation (**F**) of eNOS immunostainings of BAV and TAV tunica intima show a reduced expression in TAV compared with the other groups. Scale bar = 100 µm. Averages are reported as percentage of α-SMA^+^, S100A4^+^ and eNOS^+^ endothelial cells ± SEM. * *p* < 0.01; ^#^
*p* < 0.01 TAV < 45 mm vs. TAV ≥45 mm.

**Figure 3 ijms-23-10848-f003:**
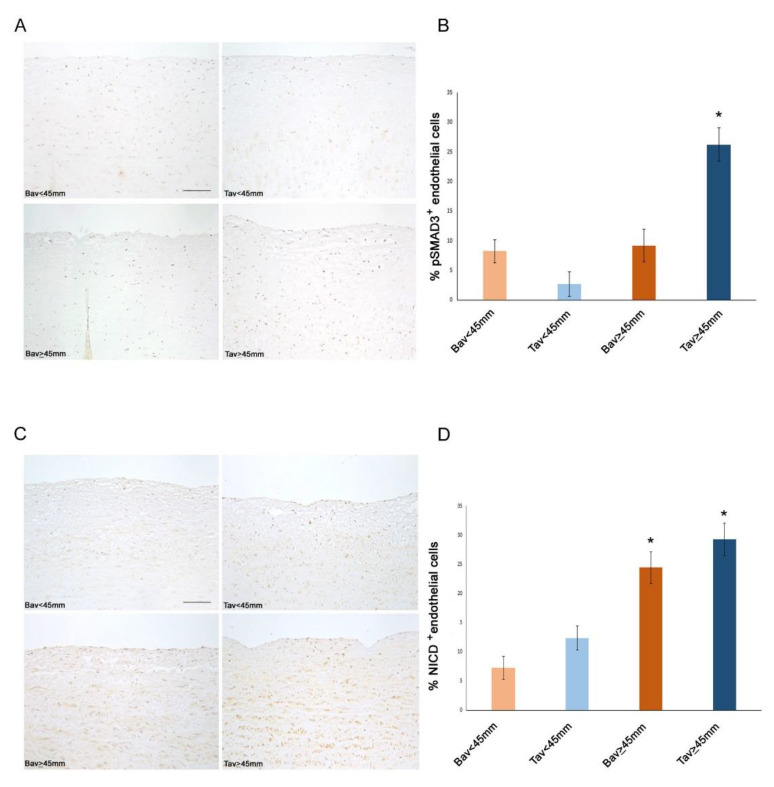
Increased pSMAD3 and NICD expression characterize TAV tunica intima with aortic diameter ≥45 mm. Representative images (**A**) and semiquantitative evaluation (**B**) of pSMAD3 immunostainings of BAV and TAV tunica intima show an increased expression in TAV with aortic diameter ≥45 mm compared with the other groups. Scale bar = 100 µm. Representative images (**C**) and semiquantitative evaluation (**D**) of NICD immunostainings of BAV and TAV tunica intima show an increased expression in BAV and TAV with aortic diameter ≥45 mm compared with the other groups. Scale bar = 100 µm. Averages are reported as percentage of pSMAD3 and NICD^+^ endothelial cells ± SEM. * *p* < 0.05.

**Figure 4 ijms-23-10848-f004:**
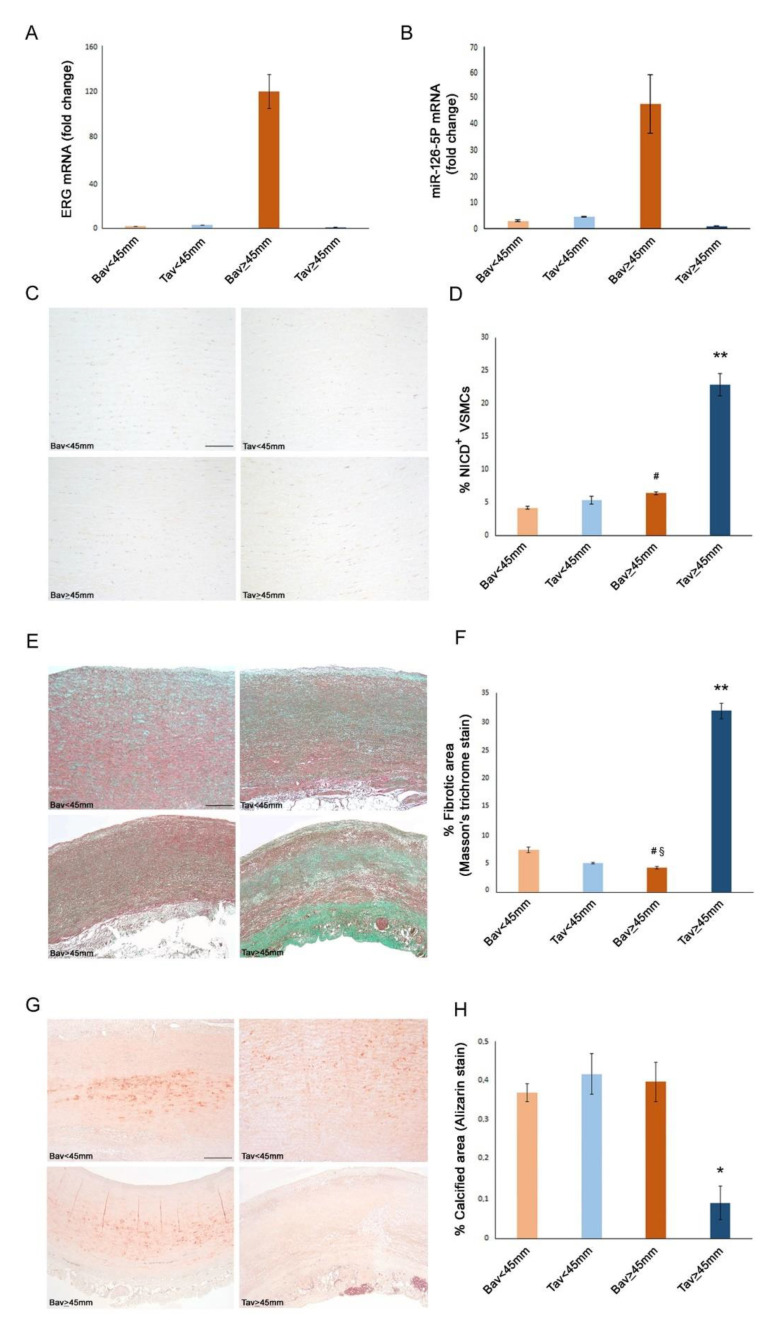
BAV tunica media with aortic diameter ≥45 mm displays increased ERG and miR-126-5P levels, whereas TAV tunica media with aortic diameter ≥45 mm shows higher fibrosis and NICD expression with lower calcification. Gene expression analysis on RNA extracted from tunica media of aortic tissue show an increased levels of ERG (**A**) and miR-126-5P (**B**) transcripts in BAV tunica media with aortic diameter ≥45 mm compared with the other groups. Changes in gene expression were calculated using the comparative ΔΔCT method. Fold change is considered significant for values >2.0 and <0.5. Representative images (**C**) and semiquantitative evaluation (**D**) of NICD immunostainings of BAV and TAV tunica media that show an increased expression in TAV with aortic diameter ≥ 45 mm compared with the other groups. Scale bar = 100 µm. Representative images (**E**) and morphometric analysis (**F**) of BAV and TAV tunica media sections stained with Masson’s trichrome that show an increased fibrotic area in TAV with aortic diameter ≥45 mm compared with the other groups. Scale bar = 500 µm. Averages are reported as percentage of NICD^+^ VSMCs ± SEM. Representative images (**G**) and morphometric analysis (**H**) of BAV and TAV tunica media sections stained with Alizarin Red that show an increased calcified area in BAV with aortic diameter ≥45 mm compared with the other groups. Scale bar = 500 µm. Averages are reported as percentage of calcified or fibrotic area ± SEM. * *p* < 0.05; ** *p* < 0.01; ^#^
*p* < 0.01 BAV < 45 mm vs. BAV ≥ 45 mm; ^§^
*p* > 0.05 TAV < 45 mm vs. BAV ≥ 45 mm.

**Figure 5 ijms-23-10848-f005:**
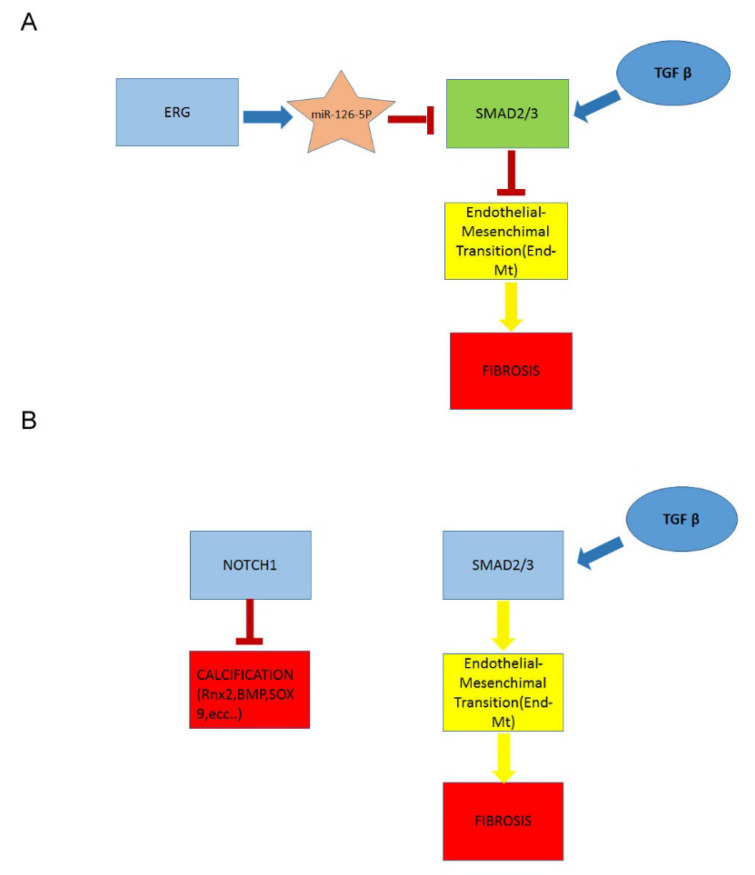
Schematic summary of altered signaling in BAV and TAV aortopathies. (**A**) Deregulated pathways in BAV aortopathy. (**B**) Deregulated pathways in TAV aortopathy.

**Table 1 ijms-23-10848-t001:** Demographic and Clinical characteristic and Histological Findings of the Study Population.

Variables	Patients (n = 20)
Demographic and Clinical Data	
Age (years)	63.7 ± 15.5
Marfan Syndrome	0 (0%)
Hypertension	15 (75%)
Diabetes	1 (5%)
Renal Failure	2 (10%)
Peripheral Vascular Disease	0 (0%)
Family History for Aneurysm	1 (5%)
Other Vascular Disease	1 (5%)
Coronary Artery Disease	9 (45%)
Valsalva Sinuses Prolapse	5 (25%)
Left Ventricular/Aortic Valve disjunction	2 (10%)
Asymmetric Dilation of Ascending Aorta	6 (30%)
Coronary Ostia Dislocation	19 (95%)
Aortic Wall Thickness	14 (70%)
Origin of the epi-aortic vessels from the aorta	18 (90%)
Ejection Fraction (%)	54.5 ± 8.4
Aortic Root Diameter	42.3 ± 5.7
Ascending Aorta Diameter	51.5 ± 7.8
**Histological Data**	
% Endothelial ERG^+^ cell/tot cells	24.4 ± 25.8
% vsmcs psmad3^+^	11.08 ± 8.97
% alfa SMA^+^ endothelial cells	23.34 ± 26.83
% S100A4^+^ endothelial cells	21.33 ± 25.33
% Fibrotic area (Masson standing)	12.23 ± 11.81
% Calcific area (Alizarin standing)	0.32 ± 0.19

**Table 2 ijms-23-10848-t002:** Demographic, clinical, histological data comparison between patients with bicuspid aortic valve (BAV) and patients with tricuspid aortic valve (TAV).

	BAV (n = 10)	TAV (n = 10)	*p*-Values
**Demographic and Clinical Data**			
Age (years)	56.1 ± 17	71.3 ± 9.5	0.024
BMI	27.8 ± 4.1	27.5 ± 4.7	0.759
Male	8(80%)	6(60%)	0.628
Caucasians	10(100%)	10(100%)	1.000
Marfan Syndrome	0 (0%)	0 (0%)	-
Hypertension	9 (90%)	6 (60%)	0.303
Diabetes	0 (0%)	1 (10%)	1.000
Renal Failure	0 (0%)	2 (20%)	0.474
Peripheral Vascular Disease	0 (0%)	0 (0%)	-
Family History for Aneurysm	0 (0%)	1 (10%)	1.000
Other Vascular Disease	1 (10%)	0 (0%)	1.000
Coronary Artery Disease	4 (40%)	5 (50%)	1.000
Valsalva Sinuses Prolapse	1 (10%)	4 (40%)	0.303
Left Ventricular/Aortic Valve disjunction	0 (0%)	2 (20%)	0.474
Asymmetric Dilation of Ascending Aorta	3 (30%)	3 (30%)	1.000
Coronary Ostia Dislocation	10 (100%)	9 (90%)	1.000
Aortic Wall Thickness	8 (80%)	6 (60%)	0.628
Origin of the epiaortic vassels from the ascending aorta	10 (100%)	8 (80%)	0.474
Ejection Fraction (%)	53.6 ± 8.6	55.4 ± 8.7	0.646
Aortic Root Diameter	42.6 ± 6.1	41.9 ± 5.6	0.792
Ascending Aorta Diameter	50.2 ± 6.8	52.7 ± 8.9	0.489
**Histological Data**			
% Endothelial ERG^+^ cell/tot cells	39.8 ± 29	9 ± 6.2	0.0082
% Vsmcs psmad3^+^	8.7 ± 3.9	13.4 ± 11.9	0.2581
% alfa SMA^+^ endothelial cells	11.5 ± 9	35.2 ± 33.6	0.0557
% S100A4^+^ endothelial cells	6.6 ± 2	36 ± 29.5	0.0117
% Fibrotic area (Masson staining)	5.9 ± 1.8	18.5 ± 14.3	0.0212
% Calcific area (Alizarin staining)	0.353 ± 0.178	0.277 ± 0.208	0.3918
% Notch^+^ Endothelial cells	13.9 ± 10.1	18.6 ± 10.4	0.3146
% Vsmcs Notch^+^ cells	5.27 ± 1.23	14.1 ± 9.6	0.0174
% eNOS^+^ cells	38.9 ± 8.4	4.9 ± 4.06	<0.001

**Table 3 ijms-23-10848-t003:** Comparison between BAV patients with an ascending aorta ≥ 45 mm (**Group 1**), BAV patients with an ascending aorta < 45 mm (**Group 2**), TAV patients with an ascending aorta ≥ 45 mm (**Group 3**), TAV patients with an ascending aorta < 45 mm (**Group 4**).

Variables	Group 1	Group 2	Group 3	Group 4	*p*-Value	*p*-Value Group 1 vs. Group 2	*p*-Value Group 1 vs. Group 3	*p*-Value Group 1 vs. Group 4	*p*-Value Group 2 vs. Group 3	*p*-Value Group 2 vs. Group 4	*p*-Value Group3 vs. Group 4	
Demographic and Clinical Data												
Age (years)	62.6 ± 15.1	49.6 ± 17.8	66.6 ± 7.7	76 ± 9.5	0.042	0.830	1.000	0.764	0.349	0.036	1.000	a
Marfan Syndrome	0 (0%)	0 (0%)	0 (0%)	0 (0%)	-							
Hypertension	4 (80%)	5 (100%)	2 (40%)	4 (80%)	0.291							
Diabetes	0 (0%)	0 (0%)	0 (0%)	1 (20%)	1.000							
Renal Failure	0 (0%)	0 (0%)	2 (40%)	0 (0%)	0.211							
Peripheral Vascular Disease	0 (0%)	0 (0%)	0 (0%)	0 (0%)	-							
Family History for Aneurysm	0 (0%)	0 (0%)	0 (0%)	1 (20%)	1.000							
Other Vascular Disease	1 (20%)	0 (0%)	0 (0%)	0 (0%)	1.000							
Coronary Artery Disease	1 (20%)	3 (60%)	3 (60%)	2 (40%)	0.762							
Valsalva Sinuses Prolapse	0 (0%)	1 (20%)	0 (0%)	4 (80%)	0.020							
Left Ventricl/Aortic Valve disjuction	0 (0%)	0 (0%)	0 (0%)	2 (40%)	0.211							
Asymmetric Dilation of Ascending Aorta	0 (0%)	3 (60%)	0 (0%)	3 (60%)	0.033							
Coronary Ostia Dislocation	5 (100%)	5 (100%)	4 (80%)	5 (100%)	1.000							
Aortic Wall Thickness	4 (80%)	4 (80%)	3 (60%)	3 (60%)	1.000							
Origin of the epiaortic vassels from the aorta	5 (100%)	5 (100%)	5 (100%)	3 (60%)	0.211							
Ejection Franction (%)	53.6 ± 11.8	53.6 ± 5	51.6 ± 9.1	59.2 ± 7.1	0.555							
Aortic Root Diameter	38.2 ± 5.8	47 ± 1.4	39.6 ± 5.7	44.2 ± 5	0.037	0.064	1.000	0.398	0.163	1.000	0.902	a
Ascendinng Aorta Diameter	56.2 ± 2.8	44.2 ± 2.4	60 ± 6.3	45.4 ± 2.5	<0.001	<0.001	0.827	0.002	<0.001	1.000	<0.001	a
**Histological Data**												
% Endotheliali ERG^+^ cell/tot cells	66.1 ± 11.1	13.5 ± 5.7	6.6 ± 3.7	11.5 ± 7.6	<0.001	<0.001	<0.001	<0.001	1.000	1.000	1.000	a
% vsmcs psmad3^+^	9.19 ± 5.03	8.25 ± 2.81	23.05 ± 9.17	3.84 ± 2.43	0.008	1.000	0.184	0.363	0.184	0.363	0.002	k
% alfa SMA^+^ endothelial cells	9.87 ± 5.98	13.2 ± 11.8	65.6 ± 14.6	4.66 ± 1.78	0.006	1.000	0.049	0.930	0.085	0.657	0.002	k
% S100A4^+^ endothelial cells	7.77 ± 1.51	5.5 ± 1.83	63.44 ± 8.03	8.62 ± 3.94	0.005	0.599	0.074	1.000	0.001	0.599	0.074	k
% Fibrotic area (Masson staining)	4.38 ± 0.51	7.52 ± 1.11	31.87 ± 3.35	5.16 ± 0.34	0.001	0.031	<0.001	0.785	0.544	0.447	0.016	k
% Calcific area (Alizarin staining)	0.34 ± 0.16	0.37 ± 0.21	0.15 ± 0.22	0.41 ± 0.09	0.144							
% Notch^+^ Endothelial cells	20.5 ± 10.1	7.3 ± 4.4	24.8 ± 11.1	12.4 ± 4.6	0.042	0.092	<0.001	<0.001	<0.001	<0.001	0.249	a
% vsmcs notch^+^ cells	6.4 ± 0.4	4.2 ± 0.4	22.9 ± 3.7	5.3 ± 1.4	0.002	0.098	0.326	0.855	0.001	0.855	0.023	k
% eNOS^+^ cells	43.4 ± 6.5	34.4 ± 8.1	1.3 ± 0.5	8.6 ± 1.6	0.001	1.000	<0.001	0.042	0.012	0.363	0.544	k

a = by ANOVA test; k = Kruskall Wallis test.

## Data Availability

Not applicable.

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
