# Peer review of "The Endothelial Transcription Factor ERG Mediates a Differential Role in the Aneurysmatic Ascending Aorta with Bicuspid or Tricuspid Aorta Valve: A Preliminary Study"

_ijms, 2022, doi:10.3390/ijms231810848_

Round 1
Reviewer 1 Report
In the present study Calogera Pisano et al. examined the involvement of ERG-related pathways in the differential progression of disease in aortic tissues from 21 patients having a bicuspid aorta valve (BAV) or tricuspid aorta valve (TAV) with or without ascending aorta aneurysms (AAA). The authors identified ERG as a novel endothelial-specific regulator of TGF-β-SMAD, Notch, and NO pathways, by modulating a differential fibrotic or calcified AAA progression in BAV and TAV aortas. Aortic calcification was indeed correlated to different ERG expression, which appeared to be under control of Notch signalling. The latter, when increased, was associated with progression versus severe complications, i.e., dissection or rupture. On the other hand, in TAV aneurysmatic aortas, ERG appeared to modulate aortic fibrosis. Therefore, the authors proposed that ERG may represent a sensitive tissue biomarker to monitor AAA progression and a target to develop therapeutic strategies and influence surgical procedures.
The study is very well designed and technically sound. In particular, the immunohistochemistry is of high quality and very convincing. The data are original and the interpretation of the findings very likely to be translated into reality through further mechanistic studies.
The study has the natural limitation of being centred on human samples, which prevents per its very nature any mechanistic evaluation of the detected results. The latter does not lessen the importance of the findings but it would be fair to acknowledge this in the discussion section.
Author Response
1.Author comment. We thank the Reviewer 1 for her/his positive evaluation of our study and results obtained. As for his/her suggestion to better emphasize the relevance of our data in the discussion, we have modified it as required.

Reviewer 2 Report
The authors provide evidence that ERG mRNA and protein are increased in endothelium for the bicuspid aortic valve in those patients with aortic diameter > 45 mm, while the expression of proteins downstream of ERG are lower in this group than in the case of tricuspid aortic valve in those patients with aortic diameter > 45 mm. The authors attribute this to differential interactions between ERG and TGFb. While the correlation coefficient r between miR-126-5P and ERG gene levels is significant, r2 is only 0.0841 suggesting a very weak relationship. Given this weak relationship, the authors don’t provide clear evidence that miR-126-5P is downregulating the End-Mt. Further, no information is provided on TGFb levels. The result is an intriguing association. As a result, the Discussion should be more tempered to indicate only what the data show and what is needed to establish a more mechanistic relationship.
Specific Comments
1. Provide a rationale for choosing a cutoff aortic diameter of 45 mm.
2. For Figures 1C, 2A, C, E, 3A,C, 4C, E, G provide a scale bar.
3. Staining for aSMA and S100A4 (Figure 2A, C), pSMAD3 and NICD (Figure 3A, C) is weak and staining of endothelial cells is not discernable. Provide higher power views of endothelium to clearly show the trends.
4. Table 2. Provide information on the sex, race and BMI of patients.
5. p. 2, lines 53-56. This sentence is unclear “In addition, we have also demonstrated that such endothelial alterations in BAV are significantly associated with an increased VSMC apoptosis evocated by an augmented expression of molecules, such as metalloproteinases (MMPs), able to cause elastin fibres and to cause calcification.” It is not clear whether the phrases “able to cause elastin fibres and to cause calcification” modify metalloproteinases or refer to molecules produced by apoptotic VSMCs.
6. In Table 3, unclear to which group(s) the p value refers. It seems to be different groups, depending on the variable. Ideally ANOVA should be performed followed by a post hoc test to identify which groups differ.
7. Check spelling throughout the document.
Author Response
The authors provide evidence that ERG mRNA and protein are increased in endothelium for the bicuspid aortic valve in those patients with aortic diameter > 45 mm, while the expression of proteins downstream of ERG are lower in this group than in the case of tricuspid aortic valve in those patients with aortic diameter > 45 mm. The authors attribute this to differential interactions between ERG and TGFb. While the correlation coefficient r between miR-126-5P and ERG gene levels is significant, r2 is only 0.0841 suggesting a very weak relationship. Given this weak relationship, the authors don’t provide clear evidence that miR-126-5P is downregulating the End-Mt. Further, no information is provided on TGFb levels. The result is an intriguing association. As a result, the Discussion should be more tempered to indicate only what the data show and what is needed to establish a more mechanistic relationship.
1.Author comment. We thank the Reviewer 1 for her/his positive evaluation of our study and results obtained. As for his/her suggestion to better emphasize the relevance of our data in the discussion, we have modified it as required.
Specific Comments
- Provide a rationale for choosing a cutoff aortic diameter of 45 mm.
2.Author comment. In the grouping of patients, we have chosen a cut-off diameter of 45 mm because is the diameter advocated in the “2021 ESC/EACTS Guidelines for the Management of Valvular Heart Disease” for ascending aorta replacement in patients with aortic valve diseases (severe stenosis or regurgitation). Patients with an ascending aorta <45 mm underwent ascending aorta replacement only in presence of specific intraoperative findings significant coronary ostia dislocation, aortic wall thickness, left ventricle/aortic valve disjunction with evidence of cardiac muscle in transparency at the level of the right/non coronary sinus, asymmetric dilatation of Valsalva sinus/sinuses
- For Figures 1C, 2A, C, E, 3A,C, 4C, E, G provide a scale bar.
3.Author comment. As suggested, we provided a scale bar for Figures 1C, 2A, C, E, 3A,C, 4C, E, G.
- Staining for aSMA and S100A4 (Figure 2A, C), pSMAD3 and NICD (Figure 3A, C) is weak and staining of endothelial cells is not discernable. Provide higher power views of endothelium to clearly show the trends.
4.Author comment. As suggested, we replaced images in the Figure 2A, C and Figure 3A, C with others at higher magnification for a better visualization of the staining.
- Table 2. Provide information on the sex, race and BMI of patients.
5.Author comment As per your request I added in TABLE 2 information on the sex, race and BMI
- p. 2, lines 53-56. This sentence is unclear “In addition, we have also demonstrated that such endothelial alterations in BAV are significantly associated with an increased VSMC apoptosis evocated by an augmented expression of molecules, such as metalloproteinases (MMPs), able to cause elastin fibres and to cause calcification.” It is not clear whether the phrases “able to cause elastin fibres and to cause calcification” modify metalloproteinases or refer to molecules produced by apoptotic VSMCs.
6.Author comment. As your suggestion, the sentence has been modified.
- In Table 3, unclear to which group(s) the p value refers. It seems to be different groups, depending on the variable. Ideally ANOVA should be performed followed by a post hoc test to identify which groups differ.
7.Author comment As per your request I correctly modified the TABLE 3. I specified which group I compered according the single variable. One-way ANOVA or Kruskal-Wallis ‘s test followed by Bonferroni correction or Dunn test was applied to compare more than two groups.
- Check spelling throughout the document.
8.Author comment. We thanks the Reviewer for his/her suggestion

Reviewer 3 Report
In the current paper, the authors describe tehri findings regarding the potential role of ERG in the progression of AAA in TAV and BAV. While this is an interesting proposition, there are several issues that fail/need to be addressed.
1. In table 2, there are several comparisons with few to no cases per groups, yet p values are provided as if statistically sound analysis is possible. I doubt that this is the case.
2. How exactly was the % ERG+ cells out of the total number of endothelial cells determined? Especially, how was the total number of endothelial cells counted?
3. In common pathology practice, ERG is used as a marker for endothelial cells, because these tend to be invariably positive. How do they explain the absence of ERG in a part of the endothelial cell population in this study?
4. The immunohistochemical micrographs in fig. 1-4 are not interpretable. Better contrast is required and a higher magnification. Was a counterstain used? If not, how were IHC negative cells recognized?
5. Were known genetic causes for AAA adequately excluded? How was this done?
Author Response
Reviewer#3
In the current paper, the authors describe tehri findings regarding the potential role of ERG in the progression of AAA in TAV and BAV. While this is an interesting proposition, there are several issues that fail/need to be addressed.
1.Author comment. We thank the Reviewer 1 for her/his positive evaluation of our study and results obtained.
- In table 2, there are several comparisons with few to no cases per groups, yet p values are provided as if statistically sound analysis is possible. I doubt that this is the case.
- Author comment. In TABLE 2, the p-values were calculated using Fisher’s exact test and not the chi-square as reported erroneously in the paragraph relating to statistical analysis which has now been correctly modified. Fisher’s exact test can also be applied in the presence of a zero. However is true that there are few comorbidities in general regardless of the subdivision into BAV and TAV, for example there are 2 cases of renal failure in all.
- How exactly was the % ERG+ cells out of the total number of endothelial cells determined? Especially, how was the total number of endothelial cells counted?
- Author comment. For the immunohistochemical evaluation, the average number of positive cells out of the total number/ high power field (at 40x magnification) was calculated by counting positive and negative cells under a light microscope (Nikon Eclipse E600) along the entire length of the aortic tissue sample, for each patient, and reported as the group average + SEM (Standard error of mean). This information was reported in the revised manuscript “see Materials and Methods Section”.
- In common pathology practice, ERG is used as a marker for endothelial cells, because these tend to be invariably positive. How do they explain the absence of ERG in a part of the endothelial cell population in this study?
- Author comment. We analyzed aortic tissue samples from unhealthy patients suffering from aortic dilatation (both TAV and BAV) and BAV, characterized by the endothelial-mesenchymal transition (endothelial cells that acquire a mesenchymal phenotype) and ERG down-regulation (Nao Naga Plos genetics 2018). Although the expression of Erg is low in the intima of dilated BAV (<45mm) and TAV aortas, a small percentage of endothelial cells is still present as reported in the Figure 1C-D. For a better interpretation of the results, we replaced images in the Figure 1 with others at higher magnification.
- The immunohistochemical micrographs in fig. 1-4 are not interpretable. Better contrast is required and a higher magnification. Was a counterstain used? If not, how were IHC negative cells recognized?
- Author comment. As suggested, we replaced images in the Figure 1-4 with others at higher magnification for a better visualization of the staining and the contrast (Haematoxylin-eosin).
- Were known genetic causes for AAA adequately excluded? How was this done?
- Author comment Our Cardiac Surgery Unit is the “Reference Center of Rare Diseases and Marfan Syndrome” in Lazio. All patients with ascending aortic aneurysm undergo clinical (Ghent Criteria) and genetic screening in order to exclude aneurysm related to familial syndrome (Marfan Syndrome, Ehlers Danlos, Loeys Diets).